# Sustainable Innovations in the Food Industry through Artificial Intelligence and Big Data Analytics

Saurabh Sharma [1], Vijay Kumar Gahlawat [1,*], Kumar Rahul [1], Rahul S Mor [2] and Mohit Malik [1]

1   Department of Basic and Applied Sciences, National Institute of Food Technology Entrepreneurship and Management, Kundli, Sonepat 131028, India; saurabhsharma_25@yahoo.com (S.S.); kumarrahul.niftem@gmail.com (K.R.); mohitmalik488@gmail.com (M.M.)
2   Department of Food Engineering, National Institute of Food Technology Entrepreneurship and Management, Kundli, Sonepat 131028, India; dr.rahulmor@gmail.com
*   Correspondence: drvijay.niftem@gmail.com; Tel.: +91-130-2281250

**Abstract:** The agri-food sector is an endless source of expansion for nourishing a vast population, but there is a considerable need to develop high-standard procedures through intelligent and innovative technologies, such as artificial intelligence (AI) and big data. This paper addresses the research concerning AI and big data analytics in the food industry, including machine learning, artificial neural networks (ANNs), and various algorithms. Logistics, supply chain, marketing, and production patterns are covered along with food sub-sector applications for artificial intelligence techniques. It is found that utilization of AI techniques and the intelligent optimization algorithm also leads to significant process and production management. Thus, digital technologies are a boon for the food industry, where AI and big data have enabled us to achieve optimum results in realtime.

**Keywords:** artificial intelligence (AI); big data; agri-food; machine learning (ML); artificial neural networks (ANN); algorithms





## 1. Introduction

Good-quality food is a fundamental requirement, and the principal aspect of any food is food safety, an integral part of food quality. Food shortage has already been a critical factor due to the tremendous population explosion and various socioeconomic factors [1]. This enormous change in population from three billion to more than six in the last five decades has created a massive demand in food consumption [2]. By 2050, there will be an approximate global increase in population by 30%, followed by the food crisis and impactful stress for increasing the food production up to 70% [3,4]. Agriculture is witnessing more diversification, causing a massive impact on non-renewable resources, which is predicted to deplete in a few decades. This sector continuously requires a proper input resource, strict monitoring, and a consumer-friendly supply chain strategy. Global warming is a significant threat due to $CO_2$ emission and deforestation, mainly because of the high utilization of resources. Food insecurity may be on the rise due to new environmental conditions adaption, which can create scarcity of resources in large areas of many countries.

The depletion of resources due to overexploitation will result in conversion to new conformation in environmental conditions, which undoubtedly becomes an impractical process in the future scenario. It leads to food insecurity, economic downfall, increased recession, hiked prices of essential goods and agricultural raw materials, human and animal epidemics, pandemics, non-reliability in farming approach, and risks [3,5,6]. Finally, food procurement remains a highlighting topic in the demand–supply chain. Selecting a practical approach among sustainable modern practices will produce better results in retaining efficiency while satisfying demand. Novel crop manufacturing and processing have been made possible due to new advances. To meet consumer demand and ensure fast production, the food industry introduced a limited number of food processing techniques.

Will these inventions be able to feed the rapidly expanding population while avoiding the inevitable? With rising demand and correspondingly rising technological advances such as AI and big data analytics (BDA), it seems probable [7].

Artificial intelligence (AI) is considered a growth ladder that influences scientists and governments, including conventional political and economic approaches to face these challenges [3]. Supporting and nurturing the increasing population through providing nutritious and safe food is quite a demanding and complex quest for the food industry. Food products are subject to numerous quality processing assessments, a more labor-intensive and cumbersome task that lacks well-equipped calibrated equipment, validation methods, and trained personnel. Accidents due to food security are rising globally, so consumers are paying more attention to food safety problems.

Forecasting hazards, risk assessment and prevention related to food safety require a simple and efficient heuristic prediction approach such as BDA to mitigate the risks. Researchers stated that "Big data is high volume, high velocity, and high variety information assets that require new processing forms to enhanced decision making, insight discovery, and process optimization" [8]. Collaborative efforts of public and private sector players are necessary to identify food safety outbreaks followed by tracing of potential causes [9]. There is a need to develop high-standard grade procedures by the food industries that are more reliable to control product quality. However, the industry possesses a varied action plan that is non-linear, corrected by an automatic and reliable approach such as AI and traceability. AI describes computational methodology depicting mental competency and an intelligent solution to varied food industry problems, leading to food security traceability [10].

Automation helps the food processing lines, and a significant amount of data is collected, stored, analyzed for further improvements in the food supply chain, and used for risk assessments. However, implementing such technologies attract hackers and creates a new risk to the industry. A study was conducted to quantify present computer science competency, exposure to food science, and familiarity with students' cybersecurity issues in food science and related fields [11]. BDA and associated technologies enhance the performance through the internet of things (IoT) for food safety and supply chain traceability to solve food security issues [12]. RFID-based transparency for implementing the rights and regulations also plays a crucial role in ensuring food security [13]. Modern digitalization generates massive data at a revolutionary pace for optimized decisions to deal with the rising problems of agricultural produce, explore the complex agrarian potential and monitor the machine functioning. Various companies and farmers could use the application of big data by analyzing and extracting productive value information from it, thus enhancing their effectiveness and opening a new gate for smart farming [4].

By gaining the advantage of artificial intelligence and BDA at every segment of linkage in the value chain addition, i.e., 'farm to fork,' food industry can fuel up the digital capability for sustainable value creation. It will also lead to food fraud prediction using data from the Rapid Alert System for Food and Feed [14]. Some AI developments in the supply chain are AI-based demand forecasting, risk management, resilience, transportation, supplier selection, inventory management, etc. [15]. Data collection and implementation can help use resources more efficiently and responsibly, improve decision allocation, and execute a circular-economy approach in the food chain. Automation and intelligent systems can also be used to select appropriate chemicals for food safety, and multitasking robots further assist in making it fast with maintained quality [16].

For sustainability, the agri-food supply chain needs to find the target areas to improve logistics [17]. The application of Industry 4.0 in modern manufacturing industries is increasingly competitive, imperative but ignored by small and medium enterprises. Such developed technologies benefit manufacturers with legacy infrastructure, small matters by providing a reference system to apply Industry 4.0 approaches in the food industry [18–21]. The authors conducted a study titled "Innovation Potentials and Pathways Merging AI, CPS, and IoT" and indicate that the human factor and its artificial collaboration dimensions

and human perspective will be critical in AI technology application in logistics [22]. Machine learning found its application in the agriculture supply chain, divided into various phases: pre-production, production, processing, and distribution [18]. Big data-driven supply chain performance measurement system results in a robust, sustainable performance in an organization [23–25].

This paper is concerned about the findings and research of the developed automation technology governing AI and BDA in the food sector. Additionally, it highlights the present technologies available in the different food processing industries. There is an urgent need to fill the void of all food and agriculture-related problems and optimize the existing processing methods available with the future sustaining automation technology. The main advantage of the assemblage of the automated technologies and intelligent systems covers reduction in error and enhanced accuracy, decision-making process, improved forecasting, increased efficiency, reduced time consumption, and cost. The rest of the paper is ordered as follows. Section 2 introduces the methodology; Section 3 presents the scope, and Section 4 gives the findings part. Section 5 contains AI and BDA applications in the food industry, followed by the implications and limitations in Section 6, and Section 7 offers the conclusions.

## 2. Review Methodology

A systematic methodology is followed to collect data and to perform analysis. In the current study, we did a systematic literature review, adopting the five-phase process given by Denyer and Tranfield (2009). In the second phase, a pilot search is performed to understand the current developments in the area better. We collected this research data from various databases such as Scopus, Science Direct, Wiley online library, Taylor & Francis, etc., with particular keywords, titles, and abstracts, including a specific period. In the third phase, we selected the relevant articles on the theme. Scopus and Web-of-Science are preferred because of their exhaustive databases [26]. The last two phases consist of the analytical part and findings as well as results. We selected the English language-based articles, and Mendeley is used to arrange the references.

## 3. Scope

Research in AI is focused on developing computational approaches to intelligent behavior to make machines more useful and intelligent [27]. Ability to learn, think, and make decisions like humans can be made possible through artificial intelligence. Various other terminologies like machine learning, deep learning, and data science are considered the subsets of AI. Machine learning prepares statistical tools to investigate the data, which can be past labeled data or clustering algorithms based on data similarity. Deep understanding provides architecture or a platform to develop multi neural networks as a tool for imitating the human brain in ANNs, Convolutional neural networks, and recurrent neural networks. The ANNs are concerned with a problem statement in numbers, and convolutional neural networks take input in pictures or images. Time-series-related issues can be dealt with recurrent neural networks.

Artificial intelligence (AI) has vast applications, including the medical and health sector, marketing and stock trading, robotics, remote sensing, heavy Industries, transportation, telecommunication, aviation, scientific discovery, and virtual games. Informatics deals with analyzing big data generated to support risk assessment, prevention, and mitigation programs to optimize food safety outcomes [9]. The current application of blockchain technology in the agri-food value chain are traceability, food manufacturing, sustainable water management, and agri-food value chain information security [28]. Narrow intelligence provides insights about specifically defined domains, like inspecting images in various scanning processes like X-ray and MRI radiology. MRI is an effective non-invasive technique for quality assessment in a wide variety of food products. Applications of MRI in evaluating body composition and fat distribution, salt and water distribution, muscle structure, cooking and freezing processes, diaries, cereals and cookies, fruits and vegeta-

bles [29]. However, general intelligence delivers to perform the results that require human intelligence [30]. As production is one of the significant areas in the industries, researchers adopted an approach regarding the manufacturing system to standardize the shop floor operations [31]. Researchers relate generalized decisions allocating the data in feedback loops for improvisation and detect a reliable mechanism for multi-rotor crewless aerial vehicles using a machine learning approach [32]. The same may be successfully applied to the food industry production line to detect no-standard products in the machine learning technique.

Further, modern data science is significant to the scientific community and digital mapping of the consumer's perspective. Researchers described the innovative technology adopted by John Deere that provides accessibility to farmers to check concerning data for decision-making [33]. Globally, the agriculture sector will soon witness a massive transformation at all levels by demonstrating the application of AI. The value of AI in agriculture was expected to show tremendous growth from USD 432 million in 2016 to USD 2.6 billion at an observed growth rate of 22.5% CAGR estimated by market researchers [34].

Big data are used in the banking, smart agriculture, finance, healthcare, smart city, and IoT-based manufacturing sector using sensors, data retrieval devices for effective data storage and processing for future endeavors [35,36]. Big data is mainly applied in observing the concerns governing global problems like food security, safety, feasibility, and progress in efficiency, which makes the scope of big data beyond imagination in farming and includes the whole food supply chain. The connection of devices in farming and supply chain leads to real-time data evolution through hassle-free wireless internet-of-things. The data sets are process-oriented, machine-oriented, and human-sourced data to predict insights in farming and business process improvements [37]. From BDA, cloud computing, clustering, data mining has to find their way as a deep root for innovation and progress in every sector [38]. Table 1 highlights the areas where AI can be used along with the particular technology.

**Table 1.** Artificial Intelligence Techniques and Applications.

| Areas | AI Techniques |
|---|---|
| marketing | ANNs<br>genetic algorithm (GA)<br>FL/modeling<br>agent-based systems (ABSs)<br>swarm intelligence (SI)<br>tree-based model<br>general forms of AI |
| logistics | ANNs<br>ABSs<br>Data mining<br>Simulated annealing<br>Automated planning<br>Robot programming<br>Heuristics |
| production | ANNs<br>FL/modeling<br>case-based reasoning<br>GA<br>ABSs<br>data mining<br>decision trees<br>daussian<br>SI |

**Table 1.** *Cont.*

| Areas | AI Techniques |
| --- | --- |
| supply chain | ANNs |
| | FL/modeling |
| | ABSs |
| | Bayesian networks |
| | SI |
| | data mining |
| | stochastic simulation |

### 4. Findings

Development of AI programs as expert systems that occasionally requires intelligence of the human level has led to significant evolution. One such program was designed at the Stanford Research Institute in 1965 named DENDRAL, which could determine the molecular structure by analyzing chemical compounds. Later on, in the mid-1970s, another program called MYCIN was developed to diagnose bacterial infections, which was designated as the first accurate expert system [39]. In another study, researchers share a clear and pragmatic idea on using chemometrics tools in food science, focusing on the composition of chemicals and chemical markers based on food authentication concerning the effects of relevant process variables [40].

Big data analytics and artificial intelligence would ensure quick decision making. Functional impacts of BDA include: (1) Descriptive analytics to identify patterns, clustering, identify agri-food risks, benchmarking. (2) Predictive analytics to predict future events by analyzing demand forecasting, price, weather, quantity, customer behavior, etc. (3) Prescriptive analytics helps make better decisions and determine what will happen using mathematical optimization and simulation. It can be used in decision-making, automatic robotics in crop harvesting and planting, and risk management [41]. The application of BDA also predicts different issues related to logistics, workforce health, safety, vehicle conditions, security, traffic rules violations, etc., to reduce supply chain risks [42]. By employing Neural Network Analysis as a platform using Big data, both online-based customer reviews and promotional marketing strategies play a huge role in predicting product demand [43]. The authors demonstrated the utilization of BDA for recognizing the relationship between the experience and satisfaction [44]. Cultured meat, artificially grown in association with tissue engineering, is one of the most impactful notable alterations in the food sector [45].

A neuromorphic robotic system was developed for a low-cost and user-comfortable platform for various neuromorphic robotics applications, where output spikes are generated through the microcontroller after connecting sensors to the microcontroller [46]. It may have significant applications in the food industry. Machine learning emerged with big data in the agri-food industry to provide better decision support and movements using AI-based sensors working on crop management, yield prediction, disease detection, and species recognition [47]. The fast-moving consumer goods (FMCG) are expanding at an incredible pace [48]. From the corporate world of business and finance to the non-corporate world of medicine and pharmaceuticals, different sectors are associated with various intelligent systems like fuzzy logic, neural networks, and algorithms of the genetic constitution [49]. A paper reported an 'Electronic Nose' capable of detecting and characterizing odors based on a combination of fuzzy and neural techniques that differentiates accurately between various coffee blends [50]. It was found that ANNs, fuzzy logic, systems with expert decisions, updated algorithms, and their combinations were considered as the finest agents in solving drying problems [51]. Drying of food has emerged in the form of a "genetically optimized fuzzy immune proportional integral derivative controller (GOFIP)" devised and demonstrated with improved accuracy, zero overshooting, and control performance and tends to reach the final moisture content value very rapidly [52]. For evaluating energy and exergy values like efficiency, loss, and utilization, with the aid of learning algorithms and

transfer functions, a feed-forward network with two-layer and 15 neurons in the concealed layer of the fluidized bed dryer for potato cubes was designed [53].

Diseases and foodborne pathogens annually cause a vast number of illnesses and deaths. For traceability in disease outbreaks and ensuring food safety, the industry is also looking at the possibilities of blockchain technology and next-generation genomic sequencing [19]. Conventional microbiological and chemical testing methods were performed previously for quality control testing through sampling, withdrawing a random sample from a time-consuming and tedious batch [54]. Climate change, change in consumer patterns, and globalization has resulted in microbiological food safety risks concerning the fresh produce supply chain. In a report, simulation modeling techniques are employed for food safety management systems via input–output operations of enhanced assessment tools for microbial food safety, modeling packaging, and enhancing the shelf life of fresh produce [55]. Hazards generation through food mismanagement affecting health is well documented [56]. Practitioners in his finding described the utility of fingerprinting to detect food adulteration. Food frauds and disputes can be identified following the food traceability system concept and involving sensitive and accurate strategies [57].

Considering health and safety as one of the primary concerns and globalization cause difficulty in food processing chains building many hurdles due to lack of automation [58]. Introduction of Automation equipped with AI and BDA will lead to numerous productive aspects like (a) minimization of time consumption; (b) fewer chances of errors; (c) increased cleanliness; (d) enhanced food safety; (e) reduction in cost; (f) decision making; (g) sorting products; (h) managing inventory; (i) waste reduction;(j) prediction of toxicity; (k) improved supply chain and logistics; (l) product development; (m) food delivery. The use of small spectral cameras to assess food quality and authenticity has grown in popularity in the business, and attempts are ongoing to offer this capability to consumers through their smartphones.

Miso's UR Robotic Arm is one of the innovative and creative inventions which serves as a flipping tool by turning the patty through initially locating the burger and gently scooping the cake and finally flipping it, thus it has got the name 'Flippy' [59]. The best part of the flippy is that it senses patty conditions and avoids overcooking. Another invention, 'Briggo,' is often considered as a robotic barista enabling robotics, cloud computing, and modern mobile technology coupled with the particular components. Briggo provides one of the best coffee experiences segmented within 40 square feet powered by modern technology [60]. The Hello Egg. is an invention that plans the meals developed by a US-based company, RND64 keeping in mind all the dietary parameters and always paving its path in providing tutorials regarding cooking [61]. Studies mention that the world with a more hygienic meal building robotic experience 'Sally' with efficient controls keeps the ingredients fresh for a long time [62]. ANNs, Genetic algorithm (GA), Fuzzy models, Agent-based/multi-agent systems (ABSs), and data mining give accurate forecasting methods and better productions. These are the regularly used techniques in production, marketing, logistics, and supply chain [63]. Table 2 represents some AI technologies used in various sectors of the food industry.

**Table 2.** Functional AI technologies in Food sector.

| Area | Algorithm | Major Contributions | Source |
|---|---|---|---|
| AI for demand forecasting | ANN | A forecasting model was presented for retailers based on customer segmentation to improve inventory data | [64] |

**Table 2.** *Cont.*

| Area | Algorithm | Major Contributions | Source |
|---|---|---|---|
| AI for risk management and resilience | 1. ANN/SVM 2. decision tree | (1). Real-time decision model incorporating an amalgamation of grey theory and layered analytic network process (ANP) to measure several resilient strategies for risk reduction. (2). A two-stage decision support system (DSS) helps managers select mitigation strategies for supply chain risk reduction. | [65,66] |
| AI for transportation | Genetic algorithm | A novel technique presented to resolve logistics via cross-docking in the supply chain. | [67] |
| AI for supplier selection | Genetic algorithm | GA based intelligent model is proposed to solve the suppliers' performance evaluation and prioritization problems | [68] |
| AI for Quality control | X-ray detection and MRI | use of X-ray imaging for the detection of defects and contaminants in agricultural commodities | [69] |
| AI for Image Processing | 1. CNN 2. Hyperspectral imaging and PCANet | 1. To detect contamination in the food tray packaging system, this technology plays an important role. 2. Food tray sealing fault detection using hyperspectral imaging and PCA Netdata | [70,71] |

## 5. Applications of AI and BDA in the Food Sector

For many years, the food sector has made much investment in processing. With the automation, they are rapidly trying to grow their scale focusing more on supply chain and logistics, leading to consumption patterns shift and predicting the market situation, especially for the products with shorter shelf life [72]. The deployment of AI and big data has not yet fully prevailed; it has shown some differences in achieving profit compared to conventional methods and techniques. Few of the AI and big data applications are described under the following food sectors: milk and milk products, meat and meat products, fruits and vegetables, bakery, beverages, spices, etc.

### 5.1. Milk and Milk Products

Milk handling and production lead to employment, but high-quality requirements and elevation in freight costs are due to its perishable and voluminous characteristics. Accordingly, innovative dairy products with less maintenance, more shelf life, and a higher degree of hygiene have become popular. New automation is needed to accomplish the market and consumer demand, strict hygiene maintenance, and new product development. Research studies depict that the prediction of milk production yields through BDA. The milk yield of cattle is very beneficial in economic terms [73]. Accuracy in forecasts regarding products tends to gather information about the shortage, efficiency, cattle's overall health, etc. Increasing the population is among the significant factors affecting dairy supply chain. Thus, forecasting can generate an edge for the future decision-making process. Another finding has delivered that modern technologies in milk procurement and billing, optimization in product composition, packaging, supply chain integration, and traceability have accounted for accurate data implementation to reduce time and cost [74]. The AI application is also demonstrated in one of the findings [75], which showed the influence

of physiological factors such as heart rate and body temperature of cattle and various environmental factors. The data gathered using the Back Propagation Neural Network and the algorithm of genetic sequence optimization. It was found that heat stress was one of the main reasons for the downfall of milk production. To release environmental stress, sprinkling water on the body of a cattle was found helpful to control humidity, heart rate, temperature, etc. A system is established for predicting the shelf life of processed cheese through Linear layer design and multiple regression model for controlling the date of expiry of the processed cheese, taking various parameters as inputs like pH, percentage of soluble nitrogen, and numerous counts of bacteria, yeasts, and molds [76]. The determination coefficient value was high, signaling the model was quite efficient, less time-consuming, and helpful for food safety consumers. The project 'Dairy Brain' focuses on utilizing big data for decision-making in dairy farms for the nutritional fulfillment of lactating cows, predicting clinical mastitis through decision-support tools [77]. Further, it necessitates improved service quality, enhanced traceability, and reduced cost to attain optimum profits [78,79].

*5.2. Meat and Meat Products*

Despite significant advancements in robotics, the food industry needs more innovations in this direction. The authors highlight the robotic advancement in enhancing processing techniques for beef, lamb, pork, seafood, fish, etc. The UK's research on the boning of beef is automated [80]. Lamb—the upturn dressing system of sheep to remove the carcass has evolved, including men and machines. The preliminary phase of slitting and cutting and performed by man, whereas the appliance deals with the other strenuous performance stages. The process superiority deals with variability in carcass handing, lowering the difficulties of the workforce, thus improving sanitization and overall output and helps reduce workforce heads. Poultry-Earlier robotics techniques equipped with sensors were supposed to show more precision in removing body parts, leading to improved hygiene as the expectancy of meat without evisceration was greater through trimming from the carcass.

Seafood—experts say that a project 'Robofish' funded by European Union for handling the fish to place them into a beheading machine as maximum yield can be generated from the slippery and flexible fish [81]. Intelligent packaging technologies were developed to detect the spoilage of seafood under varying storage conditions [82]. Techniques like hierarchical cluster analysis (HCA), principal components analysis (PCA), and partial least squares regression analysis (PLS) were applied as experimental techniques, and PLS was further selected for spoilage marker identification.

The growth of microorganisms and enzymatic activity is mainly responsible for the spoilage of meat and its products, like secondary metabolites, which ultimately pose a serious threat to health and the economy. An adaptive fuzzy logic system and FTIR (Fourier Transform Infrared) spectroscopy with the fusion of principal microbiological analysis techniques were used to determine the various parameters responsible for beef spoilage [83]. The fuzzy logic system solved meat quality estimate, microbiological growth estimation through components like fuzzifier and defuzzifier utilizing the theory and operations based on a fuzzy set of algorithm rules. A study presents meat palatability characteristics by evaluating quantitative intramuscular fat content by applying image processing techniques. Samples of meat were taken in the form of colored images, and further classification of substances (fat, muscle, connective tissue) was done with the assistance of different optimization algorithms. The three-dimensional color space characteristics of the particular importance and its intrinsic fuzzy nature led to separating the muscle from fat based on their pixel orientation with slight variation. Automating the method with a combination of techniques originating from natural evolution includes "the Fuzzy c-Means algorithm (FCM), fuzzy clustering algorithm along with GA". The results obtained from the proposed concept of the advanced image analysis help find the approximate value of the intramuscular fat content. The image only tends to show the surface of the meat.

In the future, machine vision technology could pave the way for better quantification of meat's visual appearance [84]. The authors explain the usage of sensors in KAMINA electronic nose (e-nose) to evaluate meat freshness depending on metal oxide sensor (MOS) microarray and linear discriminant analysis (LDA). E-nose is utilized to check whether the same supplier prepares the fresh meat or not through a single or multiple exposures and for determining the early decay of meat samples kept at different temperatures using LDA [85]. Authors describe that the effectiveness of cloning human behavior, AI utilized machine learning algorithms for grading quality assessment of bovine carcass. Being a task of human expertise, it is difficult to imitate by machine. However, AI made it quite possible for the quality of the meat free from human errors and could verify human expertise with 24 h availability [86].

### 5.3. Fruits and Vegetables

The difference between intelligent refrigeration and smart refrigeration is that smart refrigerators use SMS (Short Message Services) as a notification method of product age. These devised the intelligent refrigerator with the sensing ability and gave its content and period [87]. A smart refrigerator aims to list the products needed and as soon as it is matured effortlessly. Figure 1 indicates the flowchart of intelligent refrigeration [87].

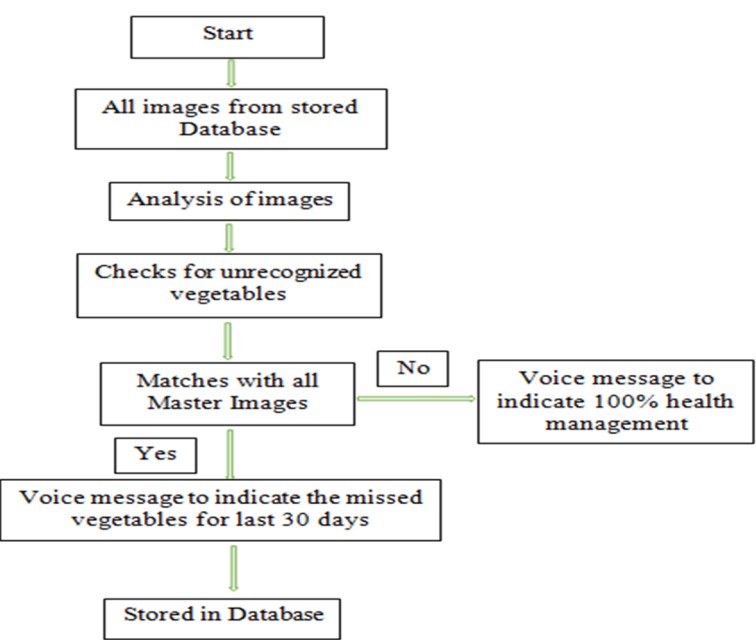

**Figure 1.** Intelligent refrigerator using artificial intelligence [87].

The intelligent refrigeration is configured into the aging algorithm, voice indicator, and image processor for its functionality. The input imagines the content that is processed and examined with the data inputs, and the output is displayed through the microprocessor in the form of signals. The result shows the age number and the age of the contents (primarily used for vegetables). The shelf life count is maintained for a maximum of 30 days. The reliability of intelligent refrigeration lies up to 96.55%. Intensive sorting of fruits and vegetables is sorting and grading the fruits and vegetables based on their maturity, weight, size, density defects, etc. Automatic visual inspection acts as a primary tool. It was designed to function using infrared colors and ultraviolet images for sorting consisting of 10 belts, with each belt consisting of a maximum of 15 fruits per second. The essential parts of automatic visual inspection consist: (1) core control unit; (2) interface panel and storage center; (3) weight sensors; (4) light sensors; (5) output unit. The authors represented a review of authentication in IoT, which provides the latest status [88,89]. A paper surveyed the grading system of various agriculture products using AI, which utilized a machine

vision strategy and included separate hardware and software components for efficient grading [85]. As the grading system was done manually, it became necessary to search for a quick and efficient technique. Table 3 shows a comparative analysis of food grading. Image processing technique, IR vision sensor, Gaussian model, Fourier-based shape separation method, Wavelet Packet Transform, Back Propagation neural network (BPNN), ANN, neuro-fuzzy model, Support Vector Machine (SVM), Multi-Attribute Decision Making (MADM), Support Vector Regression (SVR) and Fuzzy Incremental Learning (FIL) are used for efficient detection of quality products.

**Table 3.** Comparative analysis of grading.

| Fruits | Features | Technique | Accuracy |
|--------|----------|-----------|----------|
| Mango | Transition in image color | IR vision sensor and Gaussian Mixture Model | Not specified |
| Harumani mangoes | Weight, color, and shape | Fourier Based separation model | 90% |
| Cashew | Color, texture, size, and shape of cashews. | Multiresolution Wavelet transform and AI (classifier)of SVM and BPNN | 95% |
| Cherry tomato | Color, texture, shape | AI technique of SVM and KNN classifier | Not specified |
| Peanut | Color, texture, shape | AI technique of BPNN | Not specified |
| Apple | Color, texture | AI technique of FNN and SVM | 89% |
| Mango | Size, shape, weight, and surface defects of mangoes | AI technique of SVR, MADM and FIL | 87% |

Source—Survey of Grading Process for Agricultural Foods by Using Artificial Intelligence Technique.

Due to the improvement in AI, IoT, and cloud computing, the cold chain logistics linked their IT solution based on RFID, Zigbee, GPS for better quality control of its perishables. It adopts low-cost measures and acts as a troubleshooting method for load planning, route planning through AI. Eliminating the risk of temperature violence, route deviation improves the perishable commodities' quality, increasing the customer's reliability and contentment [90]. Authors discuss the digital technological developments in the context of fresh food logistics and associated operational control to enhance freshness and safety of food, reduced food wastage, and increased transportation and distribution efficiency [91].

### 5.4. Bakery

A study mentions that linear programming is an analyzing technique of mathematical background in bakeries that functions by collecting faulty materials for constructive hiking [92]. Authors express thoughts for automated control systems in the bakery industry related to the oven. With a view of upscale automation in food processing and taking advantage of a food quality sensor that converts various food properties into electrical signals, numerous quality parameters can be determined. Sensors can be grouped online and offline. These sensors are assembled in a smart oven for baking purposes. Product quality data gets stored, which in turn forecasts new values. All the necessary information from the weight of the product, time consumed, number of forms, load forecasting is generated, and the use of an intelligent control system provides adequate baking conditions, saving resources and increasing the product performance as a whole [93].

### 5.5. Beverage

The different aspects of AI, like machine learning, computer vision, robotics, and biometrics, are dependable, economical, precise, and do not require more labor effort in the beverage industry. Data in the form of images and videos collected and then refined using a

set of computerized algorithms coupled with the integration of robotics and ML modeling concepts to investigate the quality of the final products. Incorporating the sensors as a multisensory system with different automated techniques could judge or grade the effects on significant flavor parameters, taste, deformations, processing environment, etc. Moreover, robotics in the beverage industry field possesses a few applications like palletizing, processing, and packaging. The novel advancement in computer vision as a subdivision of AI catering to the values of many involving automated techniques extracting or gathering data through imitating human eye gestures. For authenticity purposes, biometrics came into the practice to determine physiological and behavioral patterns subjective to identification purposes using face recognition, body temperature, fingerprinting, and many others. It will ultimately help in the long run to access quality, consumer acceptability, decision making, monitoring, new product development, and quality control in the beverage sector. These approaches to applying AI are cost-effective, accurate, deliver profits in no time, and non-destructive [94].

Researchers revealed that ANNs act as a stool in balancing and prophesizing the various aspects governing the food sector, viz. medicine, beverages, and agriculture. As far as the brewing sector is concerned, the application of ANN is exploited to devise rapid tools for evaluating product quality and consumer acceptance. Computer vision-enabled automatic pourer, RoboBEER was used to examine numerous beer samples as triplicates for assessing different parameters related to foam generation and 15 color combinations. With the utilization of the concept of a 9-point hedonic scale, testing of the samples with 30 consumers was done based on consumer acceptability for bitterness, flavor, mouthfeel, and overall liking. Based on the above parameters, with the help of the Bayesian Regularization algorithm combined with seven neurons helped in the rapid evaluation of the beer, shows the importance of AI in the brewing industry [95].

Identification of odor and gas composition in the beverage industry was made possible through an E-nose system [96]. As aroma is mainly determined through the flavor, it especially expresses the quality of food. The system's effectiveness can quantify the volatile gaseous mixture and its mechanism of identifying all the chemical constituents. Biodeterioration is one of the major factors affecting food through numerous volatile compounds producing microorganisms. E-nose is also effective in sensing the same, which improves food preservation methods [97]. Authors study electronic nose applications to predict the type of bacteria and culture growth with a multi-layer perceptron network [98].

*5.6. Spices*

India is the leading producer of spices, and the spice industry should mandatorily undergo quality assessment. As technology is growing, the quality assessment also needs to be developed faster while considering safety. Researchers explain the implementation of computer vision and image tuning to evaluate the quality of black pepper [99]. The captured image is converted into grayscale as the information cannot be extracted from the color image. For observing the closed area on seeds, a Canny edge detection algorithm was applied on that grayscale image. With the given data, geometric parameters were deduced, and histograms were plotted. Using the gathered data, effective sorting and grading were done. The authors suggested determining the aroma and flavor of tea and spice through temperature and humidity drift compensation techniques [100]. The samples were processed by evaluating the drift coefficients of variations caused due to change in temperature and humidity and then optimized by extracting drift in E-nose response data because it may become a problem and cause data alteration. With the combination of Metal Oxide Semiconductor (MOS) sensors and various ANN techniques like Principal Component Analysis (PCA), multiple layered perceptrons (MLP), learning vector quantization (LVQ), probabilistic neural network (PNN), and radial basis functions (RBF), data can be analyzed.

## 6. Implications and Limitations

After an extensive literature review, we find some areas that can be explored further. The limitations of implementing a perfect AI technique, the impact of AI on the workforce, and challenges in AI applications may be explored. Some less popular AI techniques can be considered for improved supply chain, safety, hygiene, etc., on an applicability basis. Further, there are some challenges in AI and BDA applications, like reducing the workforce, which causes unemployment. Its implementation needs investment and also requires a skilled workforce. There are social challenges like the adoption by the workforce and a lack of decision support tools in BDA. In this study, we reviewed limited articles within a particular period, so the analysis and findings are based on the material we collected and interpreted.

## 7. Conclusions

This paper presents and elucidates the reader with a clear view of transferring or shifting from a conventional approach to the latest and novel automated system in the food sector. Although many technologies have come up to deal with the challenges that have arisen in the food sector, AI and BDA have provided legitimate platforms for experiencing the finest technology. Multidisciplinary systems like ANNs, machine learning, intelligent sensing, computer vision, fuzzy logic approach, robotics, etc., are governed with AI to evaluate various parameters depicting quality, appearance, texture, overall consumer acceptance, etc. This novel approach involved observing the pattern of data and adapting it in its workflow to bring the output that is accurate, dependable, requires less human resources, efficient, and benefits the user to predict future conditions in the long run. These techniques can be viewed as a boon for fulfilling the void in the defects still increasing in the food industry. The introduction of drone technology would slowly become another milestone in the food supply chain and logistics. Sensors are viewed as another essential tool in food preservation. Thus, AI and big data have enabled the food industry to achieve improved, optimized, and real-time outcomes.

**Author Contributions:** Conceptualization, Formal analysis, and Writing—initial draft preparation and editing: S.S. and M.M.; Review and Supervision: V.K.G., K.R. and R.S.M. All authors have read and agreed to the published version of the manuscript.

**Funding:** This research received no external funding.

**Institutional Review Board Statement:** Not applicable.

**Informed Consent Statement:** Not applicable.

**Data Availability Statement:** Not applicable.

**Acknowledgments:** We acknowledge the Department of Basic and Applied Science and Department of Food Engineering, National Institute of Food Technology Entrepreneurship and Management (NIFTEM), Kundli, Sonepat-131028, India, to support review and analytical work.

**Conflicts of Interest:** The authors declare no conflict of interest.

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
