# Peer review of "Sustainable Innovations in the Food Industry through Artificial Intelligence and Big Data Analytics"

_logistics, 2021_

Round 1

Reviewer 1 Report

Although significant work is claimed in this manuscript, but it needs minor improvements before publishing. My inputs are as follows.

  1. Latest literature should be included.
  2. The introduction section should shed light on the industry challenges and how such digital technologies can help to cope up them.
  3. Authors discussed their results well but need to relate them with similar published work, the below-mentioned papers may help to design the same.

  - Sharma, R., Kamble, S. S., Gunasekaran, A., Kumar, V., & Kumar, A. (2020). A systematic literature review on machine learning applications for sustainable agriculture supply chain performance. Computers & Operations Research, 119, 104926.

  - Baliga, R., Raut, R., & Kamble, S. (2019). The effect of motivators, supply, and lean management on sustainable supply chain management practices and performance. Benchmarking: An International Journal.

  - Kamble, S. S., & Gunasekaran, A. (2020). Big data-driven supply chain performance measurement system: a review and framework for implementation. International Journal of Production Research, 58(1), 65-86.

  - Kamble, S. S., Gunasekaran, A., & Gawankar, S. A. (2020). Achieving sustainable performance in a data-driven agriculture supply chain: A review for research and applications. International Journal of Production Economics, 219, 179-194.

  1. The manuscript needs thorough proofreading for grammatical and language errors.

    Good luck!

Author Response

Response to reviewers' attached

Reviewer 2 Report

The review manuscript entitled "Sustainable innovations in the food industry through artificial intelligence and big data analytics" needs to be revised to adequate language to appropriate format used for publications. On the abstract, please consider concise the introductory portion, then include a brief statement for the methodology, followed by results and conclusions or implications statement. For the remainder of the text, it needs to be revised to adequate words, expressions and sentences used throughout. Also, please do not use abbreviations on the beginning of sentences.

Please consider properly describing titles of figures and tables. These titles should stand alone and provide enough information for the reader to understand what that figure or table refers to.

Please consider if figures 1 and 2 are necessary- they could be described in a text format instead.

The "4.Analisys" section could probably just be part of the methodology section description.

Please check the format used for the file to aligned paragraphs properly. 

Below there are some examples of words and sentences that may not be adequate for a manuscript as they are:

  •   "an immense" - line 12
  • "as we know" - line 27
  • "every living organism" - line 28
  • "The main ... quality" - lines 29 and 30
  • "From ... investigation " - lines 675-676

Author Response

Response to reviewers' attached

Round 2

Reviewer 2 Report

Thank you for submitting the revised version. I recommend the authors to avoid using sentences with the structure below that affirms "absolute truths" for example, in the first sentence below that says for "every life form". Without getting in a complex discussion about it, I would just suggest avoiding this type of construction throughout the manuscript.

"Food performs a fundamental role in the existence of every life form. Therefore, good quality food should be consumed in the correct quantity. The principal aspect of any food is food safety which is an integral part of food quality."

Author Response

Attached for your consideration please.

This manuscript is a resubmission of an earlier submission. The following is a list of the peer review reports and author responses from that submission.